# Effects of Injection Volume and Route of Administration on Dolutegravir In Situ Forming Implant Pharmacokinetics

**DOI:** 10.3390/pharmaceutics14030615

**Published:** 2022-03-11

**Authors:** Jordan B. Joiner, Jasmine L. King, Roopali Shrivastava, Sarah Anne Howard, Mackenzie L. Cottrell, Angela D. M. Kashuba, Paul A. Dayton, Soumya Rahima Benhabbour

**Affiliations:** 1Division of Pharmacoengineering and Molecular Pharmaceutics, Eshelman School of Pharmacy, University of North Carolina at Chapel Hill, Chapel Hill, NC 27599, USA; jbjoiner@unc.edu (J.B.J.); jlking116@live.unc.edu (J.L.K.); sah@email.unc.edu (S.A.H.); 2Joint Department of Biomedical Engineering, University of North Carolina and North Carolina State University, Chapel Hill, NC 27599, USA; roopalis@email.unc.edu; 3International Center for the Advancement of Translational Science, Division of Infectious Diseases, Center for Aids Research, School of Medicine, University of North Carolina at Chapel Hill, Chapel Hill, NC 27599, USA; mlcottre@email.unc.edu (M.L.C.); akashuba@unc.edu (A.D.M.K.)

**Keywords:** long-acting, in situ, injectable, biodegradable, implants, PLGA, controlled release, drug delivery, ultrasound, pharmacokinetics

## Abstract

Due to the versatility of the in situ forming implant (ISFI) drug delivery system, it is crucial to understand the effects of formulation parameters for clinical translation. We utilized ultrasound imaging and pharmacokinetics (PK) in mice to understand the impact of administration route, injection volume, and drug loading on ISFI formation, degradation, and drug release in mice. Placebo ISFIs injected subcutaneously (SQ) with smaller volumes (40 μL) exhibited complete degradation within 30–45 days, compared to larger volumes (80 μL), which completely degraded within 45–60 days. However, all dolutegravir (DTG)-loaded ISFIs along the range of injection volumes tested (20–80 μL) were present at 90 days post-injection, suggesting that DTG can prolong ISFI degradation. Ultrasound imaging showed that intramuscular (IM) ISFIs flattened rapidly post administration compared to SQ, which coincides with the earlier T_max_ for drug-loaded IM ISFIs. All mice exhibited DTG plasma concentrations above four times the protein-adjusted 90% inhibitory concentration (PA-IC90) throughout the entire 90 days of the study. ISFI release kinetics best fit to zero order or diffusion-controlled models. When total administered dose was held constant, there was no statistical difference in drug exposure regardless of the route of administration or number of injections.

## 1. Introduction

There has been a surge in development of long-acting injectable formulations for a wide variety of sustained release applications, marked by the recently Food and Drug Administration (FDA)-approved Cabenuva for human immunodeficiency virus (HIV) treatment [1]. This formulation consists of two gluteal IM injections of cabotegravir and rilpivirine suspensions, setting a precedent for injection of multiple formulations to sustain drug release. Therefore, there is a need to understand the nuances in ISFI formation and degradation with varying injection sites and volumes. Several past and recent works have sought to understand the effects of ISFI properties on drug release using novel methods such as magnetic resonance imaging (MRI), ultrasound imaging, and ultraviolet imaging; however, to our knowledge, no works have studied the effect of ISFI formation and degradation via multiple routes of administration (including IM) and varying injection volumes [2,3,4,5,6]. This work has improved our understanding of how existing ISFI formulations work, and will serve as a guide for future development.

First described by Dunn et al., ISFIs are an injectable, sustained-release drug delivery system comprised of a drug, biodegradable polymer, and water-miscible organic solvent [7]. ISFIs have advantages over short-term delivery systems, including reduced frequency of dosing and increased patient adherence [8]. These advantages are particularly important for indications such as HIV, where incomplete adherence can lead to the development of drug resistance and loss of treatment options [9]. Drug release from ISFIs is controlled by three phases, including a 24 h burst phase, where the polymer precipitates and drug leaves with the solvent, diffusion phase, where drug diffuses through the solid ISFI, and degradation phase, where drug is released through degradation of the polymer [10]. Drug release can be tuned using several factors including excipient and drug concentrations, polymer lactic acid to glycolic acid ratio, polymer conformation, end group, and molecular weight, miscibility of the solvent in water, and drug physicochemical properties such as crystallinity, molecular weight (MW), LogP, and pKa [11,12,13]. Previously, our group demonstrated the development of removable, room-temperature stable ISFIs that released antiretroviral drugs for several months up to one year and provided HIV protection in a humanized mouse model [14,15]. Here, we selected promising ISFI formulations with dolutegravir (DTG) from this study, including 100 and 250 mg/mL DTG in 1:2 *w*/*w* poly(DL-lactide-*co*-glycolide) (PLGA): *N*-methyl-2-pyrrolidone (NMP) and investigated the effects of injection volume, number of injections, and route of administration on ISFI degradation and PK in mice to determine future administration parameters in a translational setting.

## 2. Materials and Methods

### 2.1. Materials

We purchased 50:50 Poly(DL-lactide-*co*-glycolide), ester terminated (PLGA) from LACTEL (Birmingham, AL; Cat. No. B6010-1P, Lot# A17-142, weight average MW: 27.2 kDa, intrinsic viscosity (i.v.): 0.38, polydispersity index (PDI): 1.81. *N*-methyl-2-pyrrolidone (NMP, <USP>) was received from ASHLAND (Wilmington, DE, Product Code 830697, 100%NMP). Dolutegravir (DTG) was purchased from Selleckchem (Houston, TX, #S2667). Gelucire, phosphate buffered saline (PBS), high-performance liquid chromatography (HPLC)-grade Acetonitrile and Water were purchased from Sigma Aldrich (St. Louis, MO, USA).

### 2.2. Methods

#### 2.2.1. High-Performance Liquid Chromatography (HPLC)

To quantify the residual concentration of DTG in ISFIs ex vivo, ISFIs were retrieved and flash frozen at −80 °C post euthanasia at 90 days post-ISFI administration. DTG was extracted by incubating the ISFIs in acetonitrile (ACN) overnight and quantified by HPLC analysis. A validated reverse-phase HPLC method was used to quantify the concentration of drug released in vitro from various ISFI formulations [15]. The HPLC analysis was carried out in a Thermo Finnigan Surveyor HPLC (Thermo Finnigan, San José, CA, USA) with a Photodiode Array Plus Detector, LC pump plus with auto sampler on an Inertsil, ODS-3 column (4 μm, 4.6 Å ~ 150 nm) (GL Sciences, Torrance, CA, USA) stationary phase. The column was maintained at 40 °C, with a flow rate of 1.0 mL/min with a 25 μL sample injection. A mobile phase of water:ACN 95:5 *v*/*v* and 0.1% trifluoroacetic acid (TFA) was used for DTG and read at 265 nm. A gradient method was utilized to achieve separation (0–20 min: 5%–100% ACN; 20–22 min: 100% ACN; 23–25 min: 5% ACN).

#### 2.2.2. Preparation of ISFI Formulations

PLGA was mixed with NMP at 1:2 weight ratio (*w*/*w*), and dissolved by vortex mixing at room temperature to make a homogeneous placebo formulation. For drug-loaded formulations, DTG was added at a concentration of 100 mg/mL or 250 mg/mL to the PLGA/NMP placebo formulation and mixed at 37 °C overnight to obtain a homogenous drug-loaded formulation. For the 250 mg/mL DTG formulation, Gelucire was added as a solubility enhancer for DTG at a 1:9 *w*/*w* ratio to NMP. To confirm the homogeneity of DTG in the ISFI formulation, sample aliquots (1–2 mg, *n* = 4) were collected from four different areas in the formulation, dissolved in acetonitrile (1 mL), and the drug concentration was quantified by HPLC analysis. A formulation was considered homogeneous when the average concentration in all four aliquots had a standard deviation of ≤5%.

#### 2.2.3. In Vitro Drug Release from DTG ISFIs

To understand the effect of injection volume on release of DTG from ISFIs in-vitro, we injected 40 μL (40 ± 4 mg, *n* = 4) into 200 mL of release medium (0.01 M PBS pH 7.4 with 2% Solutol) and incubated under sink conditions at 37 °C. Sink conditions are defined as in [14]. Then, 1 mL sample aliquots were collected at timepoints from 0 to 90 days and replaced with 1 mL of fresh medium. Sink conditions were maintained by completely removing the release medium and replacing with 200 mL of fresh release medium weekly. Drug concentration was quantified by HPLC using the method described in Section 2.2.1, and cumulative drug release was normalized by the total mass of drug in each depot.

#### 2.2.4. Ultrasound Imaging of ISFIs in Mice

A 60-day in vivo study was carried out to assess the effects of route of administration and injection volume on ISFI formation, shape, and degradation. B mode ultrasound imaging was performed over 60 days to measure the volume, surface area, and echogenicity of placebo ISFIs (1:2 PLGA:NMP) of varying injection volumes and routes of administration. In this study, female NOD scid gamma (NSG) mice were injected either SQ with 1 × 80 μL (*n* = 8) or 2 × 40 μL (*n* = 5) ISFIs or IM with 1 × 40 μL (*n* = 10) ISFI. Injection locations are shown in Figure 1. Mice were shaved and imaged at 24 MHz using B mode (grayscale brightness mode) ultrasound with a Vega (SonoVol, Inc., Durham, NC, USA) 3-D mouse imaging system. B mode imaging was performed at the following time points: 1 d, 2 d, 5 d, 7 d, 14 d, 21 d, 30 d, 45 d, and 60 d. Mice were anesthetized with 1.5% vaporized isoflurane in oxygen, and body temperature was maintained using a heat lamp. Mice with SQ ISFIs were imaged supine, while mice with IM ISFIs were imaged in the right lateral recumbent position. After acquisition of the B mode images, the sonographer selected a region of interest around the ISFI boundary in the tissue using SonoEQ (SonoVol, Inc., Durham, NC, USA). The sonographer performed the image segmentation of all images on *n* = 3 separate occasions to assess intrareader variability. To calculate the ISFI volume and surface area, the sonographer manually drew 2-D regions of interest (ROI) around the ISFI. In the software, intermediate slices are interpolated to count the number of voxels in the ROI and multiplied by the voxel volume.

#### 2.2.5. Statistical Analysis of Ultrasound Images

To analyze differences in in vivo ISFI volumes and surface area between groups based on imaging data, unpaired t tests (comparison between two groups) or one-way ANOVA (comparison between three groups) were performed across all timepoints. Statistical analyses were performed in GraphPad Prism 7 (GraphPad Software, Inc., La Jolla, CA, USA).

#### 2.2.6. In Vivo Pharmacokinetic Studies

A 90-day in vivo study was carried out to assess the effect of ISFI volume and number of injections on DTG PK. All experiments involving mice were carried out with an approved protocol by the University of North Carolina Animal Care and Use Committee. Female NOD scid gamma (NSG) mice, 6–8 weeks (Jackson Laboratory), were housed in a pathogen-free room. In vivo PK studies were carried out with 2 single drug ISFI formulations. Liquid ISFI drug formulations were administered subcutaneously with a 19G needle on the shaved back or hind leg of anesthetized NSG mice (Figure 1). Peripheral blood was collected from mice into capillary tubes coated with ethylenediaminetetraacetic acid (EDTA) to isolate plasma. ISFIs were harvested from *n* = 3 mice per group at days 30, 60, and 90. All samples were stored at −80 °C until analysis.

#### 2.2.7. Mass Spectrometry of Plasma Samples

Drug concentrations were measured in plasma using liquid chromatography–mass spectrometry (LC-MS/MS). Plasma samples were extracted by protein precipitation with methanol containing the stable, isotopically labeled ^13^C, d_5_-DTG as the internal standard. Following precipitation, samples were mixed 1:1 with water prior to LC-MS/MS analysis. Analytes were separated on an XTerra MS C18 (50 × 21 mm, 3.5 μm) analytical column (Waters, Milford, MA, USA) prior to detection on an API-5000 triple quadrupole mass spectrometer (AB SCIEX, Foster City, CA, USA). Calibration standards and quality control samples were within 20% of nominal values with a dynamic range of 50–50,000 ng/mL. Composite concentration-time profiles were constructed for each analyte and visually inspected to make qualitative between analyte comparisons.

#### 2.2.8. Noncompartmental Analysis of Pharmacokinetic Data

PK parameters were estimated using non-compartmental analysis in Phoenix WinNonlin version 8.3; (Certara, L.P., St. Louis, MO, USA). The log-linear trapezoidal method was used to calculate the partial area-under-the-concentration-time-curve over the 30-day study period (AUC_0–30d_). Descriptive statistics using the Phoenix WinNonlin software were performed on all pharmacokinetic estimates. Pharmacokinetic data are presented as median with interquartile range (IQR) in all treatment groups.

#### 2.2.9. Deconvolution Analysis of Pharmacokinetic Data

Deconvolution was performed in Phoenix WinNonlin version 8.3 (Certara L.P., St. Louis, MO, USA) to estimate the in vivo release rate of each ISFI treatment group based on observed plasma DTG concentrations. Unit impulse response parameters for deconvolution were derived from previously published literature by allometrically scaling the DTG half-life observed in rats (6.18 h) given a single 1 mg/kg IV bolus dose of DTG [15] using the following equation: $T_{1/2,\;mouse}=T_{1/2,\;rat}*{\left(Weight_{mouse}\;÷Weight_{rat}\right)}^{0.25}$. Rat DTG volume of distribution (100 mL/kg) was scaled assuming proportionality with body weight and used to calculate the macro constant A (or the intercept of concentration versus time at time 0). For scaling, rat and mouse weight was assumed at 0.2 kg and 0.02 kg, respectively. Loaded dose of DTG for each ISFI treatment was calculated in ng/kg units based on the first observed mouse weight or when not available (*n* = 30 mice were missing weight observations) the average weight of all mice included in the dosing studies, which was 22.7 g. To analyze the kinetics of DTG release from the ISFI in vivo we tested three models (zero order, first order, and diffusion-controlled according to the Higuchi equation) as previously described [16]. Linear regression was performed using SigmaStat v13 (Systat Software, Inc., San Jose, CA, USA) to determine the R^2^ value for each of these models.

#### 2.2.10. Statistical Analysis of Pharmacokinetic Data

To analyze differences in AUC_0–30d_ between groups a Kruskal–Wallis one-way analysis of Variance on Ranks were performed using SigmaStat v13 (Systat Software, Inc.).

## 3. Results

The goal of this work was to study the effect of injection volume, number of injections, and route of administration on the PK of promising DTG ISFIs selected from previous work [14,17]. We previously demonstrated the utility of ultrasound imaging to noninvasively assess ISFI formation and bioerosion [14]. The solidity of the ISFI can be visualized by its echogenicity in the image; high contrast indicates more solid due to PLGA precipitation and lower contrast indicates less solid due to high solvent content or PLGA degradation. In this study, B mode ultrasound (24 MHz) was utilized to image placebo ISFIs to visualize and measure the volume, surface area, and echogenicity of ISFIs with varying injection volumes and routes of administration over time. In parallel, a PK study was performed by varying injection volume of DTG ISFIs, routes of administration, and drug loading to determine how these parameters would affect plasma levels of DTG in mice. The study plan is shown in Figure 1.

### 3.1. In Vivo Imaging of Placebo ISFIs

ISFI formation and degradation play a key role in drug release [10]. We hypothesized that injection volume influences ISFI formation and degradation, and can therefore be adjusted in order to confer a desired drug release profile. Injection site has also been demonstrated to affect ISFI disposition; however, to our knowledge, a pharmacokinetic comparison between SQ and IM ISFIs has not been previously explored [3]. Therefore, we performed a longitudinal imaging study using B mode ultrasound varying ISFI injection volume and route of administration in NSG mice. Our previous work demonstrated that the 1:2 (*w*/*w*) PLGA:NMP formulation demonstrated ultra-long-acting release (>8 months) of DTG [14]; therefore, this formulation was selected to carry out the present studies. In our previous study, there was no statistically significant difference in the ultrasound-segmented volume of DTG-loaded compared to placebo ISFIs [15]; therefore, we utilized placebo ISFIs for the imaging portion of the study. In order to determine the range of injection volumes, we performed a maximum tolerated dose (MTD) study of 1:2 (*w*/*w*) PLGA:NMP ISFIs in NSG mice injected SQ and IM. We found that SQ, mice were able to tolerate an 80 μL injection (58 mg NMP), but a 100 μL injection (73 mg NMP) was lethal, presumably due to the high solvent levels. Mice were able to tolerate a 40 μL IM injection (29 mg NMP), but an 80 μL IM injection was lethal. Interestingly, this trend is opposite for humans, where larger volumes can be injected IM than SQ [18]. Based on these limitations, we injected placebo ISFIs and performed B mode ultrasound imaging according to Figure 1A.

The process of phase inversion was observed in the first two days, as seen in the 80 μL SQ ISFIs shown in Figure 2 (axial view). On day 1, there was a large, hyperechoic region, which represents the ISFI still in solution, surrounded by an area of precipitating PLGA (shown by the white arrows). On day 2, this boundary diffused further into the ISFI due to increased PLGA precipitation. In the sagittal view, only the bottom half of the ISFI was observed due to shadowing (indicating a more round, solid implant) until day 7, where the ISFI swelled due to water uptake. At day 14, the top of the ISFI was visible, which indicated that the ISFI started to flatten due to initiation of PLGA bulk degradation via hydrolysis. The ISFI were visible on B mode through 45 days, and after sacrifice on Day 60, no ISFI was found in the SQ area, demonstrating that ultrasound can be used to track the entire degradation of the ISFI.

Figure 3 depicts 40 μL SQ and IM ISFIs, which were imaged over 30 days. Due to the small size of ISFIs, it was not facile to observe the phase inversion process in the 40 μL ISFIs using B mode imaging. However, there were some notable differences between the SQ and IM ISFIs of the same size. Firstly, the boundaries of the IM ISFIs were much easier to delineate than the SQ ISFIs in both the axial and sagittal views. Secondly, the 40 μL SQ ISFIs exhibited a flattening in the sagittal view that occurred at day 14, similar to the 80 μL ISFIs in Figure 1. On the other hand, it was much easier to see the top portion of the IM ISFIs at day 1, indicating that a flattening occurred within the first 24 h of injection. This is likely due to the higher interstitial pressure of muscle versus skin in vivo [18,19,20], and correlates to differences in drug release between IM and SQ ISFIs discussed in subsequent sections.

ISFIs were segmented in all three dimensions as shown in Appendix A. The segmentation results of ISFIs generated from 1 × 80 μL SQ, 2 × 40 μL SQ, and 1 × 40 μL IM injections are shown in Figure 4. Segmentations of ultrasound images demonstrated statistically different volume measurements of ISFIs generated with 80 μL SQ injections compared to 40 μL SQ injections (Figure 4A). When administered SQ, ISFIs formed by phase inversion swelled due to water uptake through 7 days post-administration, demonstrated by a decrease in surface area within the first 7 days. At day 14 post-administration, ISFIs started to flatten as demonstrated by an increase in surface area for ISFIs generated by both 40 and 80 μL SQ injections (Figure 4B) and corroborated with ultrasound images in Figure 3A. The volume and surface area subsequently decreased as PLGA began to degrade via ester hydrolysis, with complete degradation observed between days 30 and 45 for ISFIs generated with 40 μL SQ injections and between days 45 and 60 for ISFIs generated with 80 μL SQ injections. In addition, the surface area to volume ratio is higher for the smaller ISFIs (40 μL) compared to the larger ISFIs (80 μL), which allowed for increased diffusion of drug through the ISFI (Figure 4C). As demonstrated visually in the ultrasound images (Figure 3) and by the calculated surface area (Figure 4D), IM ISFIs exhibited rapid flattening post administration compared to SQ ISFIs. Moreover, the surface area of IM ISFIs remained relatively constant through the first seven days, in contrast to the surface area of SQ ISFIs, where a decrease in surface area was observed due to swelling over the first 7 days (day 7 *p* < 0.001). However, by day 14, the SQ ISFIs flattened and showed similar surface area results compared to IM; therefore, the overall difference between the two groups was not significant. There was no statistically significant difference in the volume or surface area to volume ratio when comparing 40 μL IM versus SQ injections (Appendix A).

### 3.2. In Vivo Pharmacokinetics of Dolutegravir ISFIs

In order to correlate the effects of ISFI formation and degradation seen using ultrasound imaging, PK studies were performed with DTG-loaded ISFIs. The study design is illustrated in Figure 1B, and the individual mouse DTG plasma concentrations for each group is shown in Appendix A. All formulations maintained DTG plasma concentrations above four times the PA-IC90 (256 ng/mL) over 90 days (Figure 5). Maximum plasma concentrations (C_max_) occurred within 24 h post injection for all formulations. For several of the SQ administered ISFIs, a slight increase in DTG plasma concentration was observed at day 7 (100 mg/mL 1 × 80 μL, 2 × 40 μL and 1 × 40 μL), which was attributed to ISFI swelling resulting in greater DTG diffusion from the ISFIs (Figure 5A,B), which could be attributed to the increase in surface area observed at day 14 (Figure 4B). Furthermore, for the 1 × 80 μL 250 mg/mL DTG ISFIs, several of the mice exhibited a significant increase in DTG plasma concentration at day 14 (Figure 5C and Appendix A). We hypothesize that this is due to the DTG concentration in this formulation (250 mg/mL) being close to the saturation solubility of DTG (255 ± 4 mg/mL) in NMP and therefore exhibiting limited stability at room temperature [15]. The mice that received 1 × 80 μL 250 mg/mL DTG-ISFI injections were the last of the 250 mg/mL group to be injected (Appendix A) and potentially received unstable formulation, explaining why this group experienced increased plasma levels at day 14. When the DTG dose was increased from 8 to 20 mg, the AUC of DTG plasma levels were not significantly increased within 90 days regardless of injection number. This could lead to longer-lasting ISFIs; however, longer-term studies would be needed to confirm this hypothesis (Appendix A).

In addition to these analyses, we performed a partial and pooled non-compartmental analysis (NCA) based on the PK sampling in this study (Table 1, Appendix A). Because plasma samples were collected for all mice through day 30, this endpoint was used to conduct a partial NCA. Based on this analysis, there was no statistically significant difference in the partial AUC between treatment groups that received the same dose. Therefore, an increase in the number of injections or the route of administration did not have an effect on drug exposure during this study period. NCA (Appendix A) confirmed plasma DTG observations (Figure 6) suggesting that the route of administration has an effect on T_max_, which correlate to findings from the ultrasound imaging, explaining differences in ISFI formation and degradation in the various tissues (Figure 4D). The T_max_ for IM injections of DTG-ISFIs occurred 3-h post-injection, while the T_max_ for SQ injections occurred 1-day post-injection, as illustrated in the inlayed graphs in Figure 6. This can be further explained by the ultrasound imaging data, which showed that IM ISFIs flattened within the first day of injection, sustaining its high surface area in contrast to SQ ISFIs. SQ ISFIs exhibited a more spherical shape after injection due to swelling, showed by a decreasing surface area (Figure 4D). From this data, the disposition of the solvent is still unknown; therefore, further exploration is needed to understand whether the ISFI flattening, departure of solvent, or both, is dictating the observed difference in the onset of C_max_ between IM and SQ injections. Although a difference in C_max_ and T_max_ was observed between the IM vs. SQ administration of 4 mg DTG-ISFI, SQ administration of DTG did not differ significantly in AUC compared to IM administration, respectively (2 × 20 μL, *p* = 0.999; 1 × 40 μL, *p* = 1.000). In the case of this DTG formulation, the AUC values of SQ and IM were similar, demonstrating that both routes of administration can lead to predictable and adequate DTG levels, but IM injections could be beneficial in situations where earlier T_max_ is needed.

### 3.3. Deconvolution Analysis of Dolutegravir ISFIs

Deconvolution analysis demonstrated controlled release of DTG for up to day 90 with r^2^ values of ≥0.78 for mathematical models of zero order and diffusion-controlled release (Figure 7D) for all dosing strategies except 2 × 20 μL IM injections. Individual μg/day, cumulative μg, and fraction released for mice in each group are shown in Appendix A. The median release per day over 90 days for all formulations ranged between 15.32 and 542.04 μg/day (Figure 7A). In vitro drug release of 100 mg/mL 40 μL ISFIs was zero order between days 30 and 90, with a release per day of 20.7 μg/day (Appendix A). This falls within the range of DTG release per day for the 100 mg/mL 40 μL formulations between days 30 and 90 of 15.32–42.36 μg/day, demonstrating reasonable agreement between in vitro and in vivo estimations. It should be noted that 40 μL is the maximum volume that can be injected per implant in vitro directly into release media in a controlled manner; therefore, in vitro comparisons cannot be made with 80 μL ISFIs. Interestingly, administering the ISFI with multiple injections appeared to shift DTG release kinetics, where cumulative fraction released over time was best explained by the zero-order model (r^2^ = 0.91–0.97). In contrast, all doses administered by a single SQ injection best fit the Higuchi model of diffusion-controlled release (r^2^ = 0.87–0.89) (Figure 7D and Appendix A). We hypothesize that the higher surface area to volume ratio of multiple, smaller implants allows for zero order release. In future studies, we plan to test whether this effect holds true with larger implants in larger animal models. The fraction of remaining DTG was estimated by the deconvolution model (Figure 7C) with no apparent trend between drug loading, number of injections or total dose on fraction of drug remaining at Day 90. ISFIs were also harvested at 90 days post-injection (*n* = 3) and the remainder of drug was quantified, shown in Appendix A, where a decreasing trend for μg of DTG remaining based on decreasing injection volume was observed for the 100 mg/mL DTG formulations. However, the opposite trend was observed for the 250 mg/mL formulation, where increasing number of injections had more drug remaining. We hypothesize that this was due to the large amount of drug release of the 250 mg/mL 1 × 80 μL at Day 14 post-injection, correlating with the flattening that occurs at Day 14 in the 80 μL implants (Figure 2B and Figure 4B). The residual DTG at Day 90 was also used to calculate the fraction released to compare with deconvolution model values. The model was able to reasonably predict the quantified DTG remaining at Day 90, with the exception of the 100 mg/mL 1 × 40 μL IM group, for which the model predicts that all of the drug should be released at Day 90. However, the analytical data suggest that 39% of DTG still remained in the ISFI. The model also underestimated the amount of DTG released from the 250 mg/mL 1 × 80 and 2 × 40 μL SQ groups, which predicted a fraction of 0.44 and 0.48 of total DTG released at Day 90. The analytical values, however, determined an increased fraction of 0.998 and 0.914 DTG released. These model mismatches could be due to the fast phase-inverting properties of the IM formulations and the stability of the 250 mg/mL DTG formulation.

## 4. Discussion

We developed an ISFI capable of delivering DTG in mice four times above the PA-IC90 target concentration via the SQ and IM routes using a variety of injection volumes. Using ultrasound imaging and PK, we demonstrated that DTG-ISFIs injected via the IM route undergo phase inversion more rapidly than SQ, leading to an IM T_max_ of 3 h and SQ T_max_ of 1 day. As demonstrated from the ultrasound imaging studies, placebo ISFIs exhibited faster in vivo biodegradation than DTG ISFIs. This phenomenon is consistent with our prior findings [14] and demonstrates the effect of drug properties (LogP, pKa) on the rate of degradation of PLGA. Future studies can be conducted with hydrophilic drugs to capture whether this trend holds true for drugs with varying physicochemical properties. Administering the same drug dose over multiple ISFI injections led to similar AUC values compared with one formulation, despite the significant differences in surface area to volume ratio as observed with ultrasound imaging. Route of administration also did not have a significant effect on AUC. These effects should also be investigated in larger animal models to replicate clinically relevant doses. With the increased development of ISFIs for a variety of applications, it is crucial to understand how the route of administration, injection volume, and number of injections affect ISFI formation, degradation, and drug PK. These results can help researchers determine formulation parameters and/or route of administration to accommodate patient needs.

## Figures and Tables

**Figure 1 pharmaceutics-14-00615-f001:**
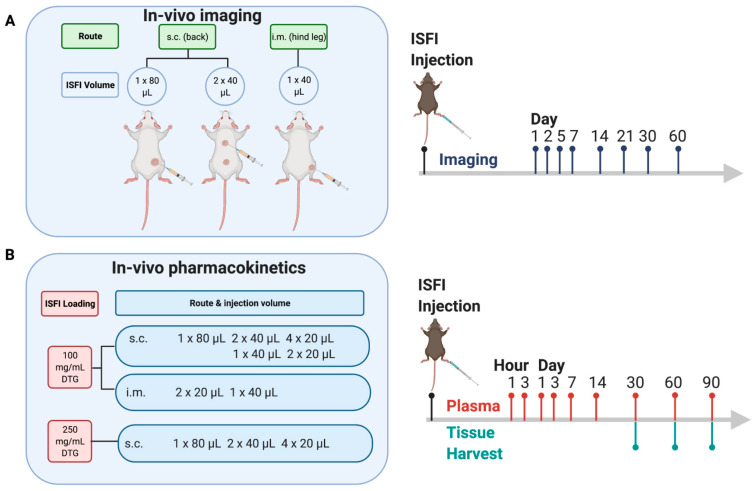
Study design to determine the effects of injection volume, number of injections and route of administration on ISFI degradation and drug release: (**A**) B mode ultrasound imaging of placebo ISFIs: SQ (1 × 80 μL, *n* = 8 mice/group and 2 × 40 μL, *n* = 5 mice/group) and IM (1 × 40 μL, *n* = 10 mice/group). (**B**) Pharmacokinetic study of 100 mg/mL and 250 mg/mL DTG ISFIs injected via SQ and IM with varying number of injections and injection volumes (*n* = 10 groups; *n* = 12 mice/group; *n* = 120 total mice).

**Figure 2 pharmaceutics-14-00615-f002:**
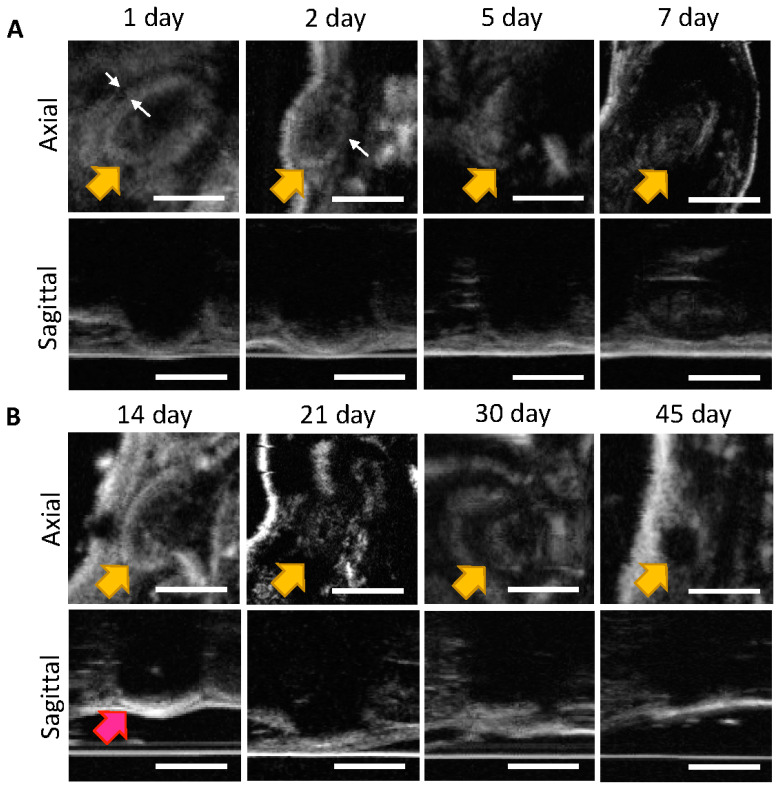
Representative B mode ultrasound images of axial and sagittal views of 80 μL placebo ISFI (1:2 *w*/*w* PLGA/NMP) ISFIs injected into female NSG mice (*n* = 8) taken over 45 days: (**A**) Days 1–7 (**B**) Days 14–45. Yellow arrows indicate location of the ISFI, white arrows indicate precipitation of PLGA, and pink arrows represent ISFI flattening. Scale bar = 5 mm.

**Figure 3 pharmaceutics-14-00615-f003:**
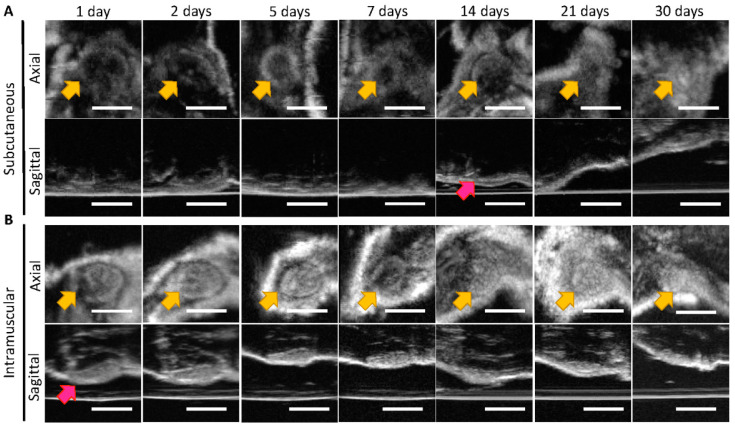
Representative B mode ultrasound images: (**A**) SQ (**B**) IM of axial and sagittal views of 40 μL placebo ISFI (1:2 *w*/*w* PLGA/NMP) ISFIs injected into female NSG mice (*n* = 5) taken over 30 days. Yellow arrows show ISFIs in the axial, and the timepoint at which the ISFI flattens is shown by the pink arrows. Scale bar = 5 mm.

**Figure 4 pharmaceutics-14-00615-f004:**
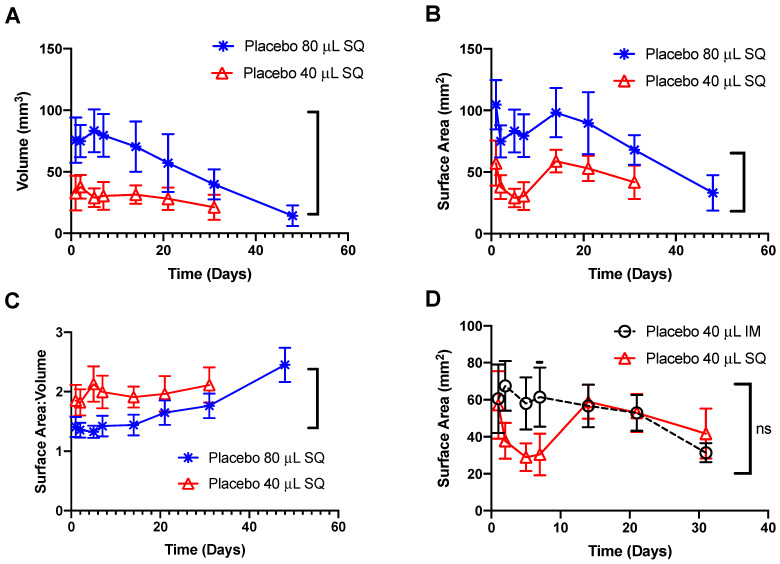
Volume and surface area of 80 μL (*n* = 8 mice/group; 1 ISFI/mouse) and 40 μL (*n* = 5 mice/group; 2 ISFIs/mouse) placebo ISFIs (1:2 *w*/*w* PLGA/NMP) injected into female NSG mice: (**A**) Volume of 80 and 40 μL SQ placebo ISFIs over 45 days (*p* = 0.0043), calculated by segmenting the ISFI boundary from multiple slices across 3 dimensions and interpolating the segmented volume. (**B**) Surface area of 80 and 40 μL SQ placebo ISFIs over 45 days (*p* = 0.0027), calculated by segmenting the ISFI boundary from multiple slices across 3 dimensions and interpolating the segmented surface area. (**C**) Surface area to volume ratio of 80 and 40 μL SQ placebo ISFIs over 45 days (*p* = 0.0281). (**D**) Surface area of SQ and IM 40 μL placebo ISFIs over 30 days (n.s.; day 7 individual *p* < 0.001). All statistics were unpaired t-tests.

**Figure 5 pharmaceutics-14-00615-f005:**
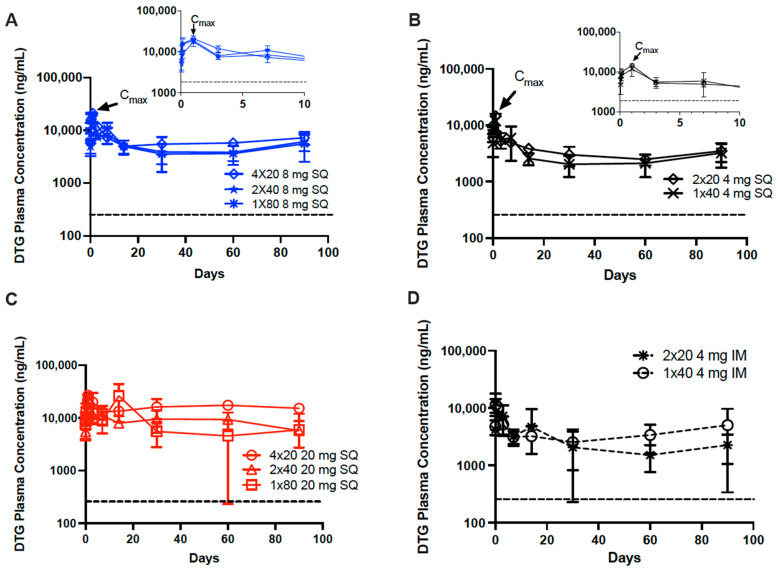
DTG plasma concentrations of 4 × 20, 2 × 40, and 1 × 80 μL DTG-loaded ISFIs (1:2 *w*/*w* PLGA/NMP) injected into female NSG mice (*n* = 10 mice/group). Each graph compares the effect of injection volume when keeping total dose constant, the black dashed line represents the 4 × PA-IC90 for DTG (256 ng/mL): (**A**) PK of 100 mg/mL DTG ISFIs given SQ with 8 mg total dose. Inset: PK through day 10 to show C_max_ at Day 1. (**B**) PK of 100 mg/mL DTG ISFIs given SQ with 4 mg total dose. Inset: PK through day 10 to show C_max_ at Day 1. (**C**) PK of 250 mg/mL DTG ISFIs given SQ with 20 mg total dose. (**D**) PK of 100 mg/mL DTG ISFIs given IM with 4 mg total dose.

**Figure 6 pharmaceutics-14-00615-f006:**
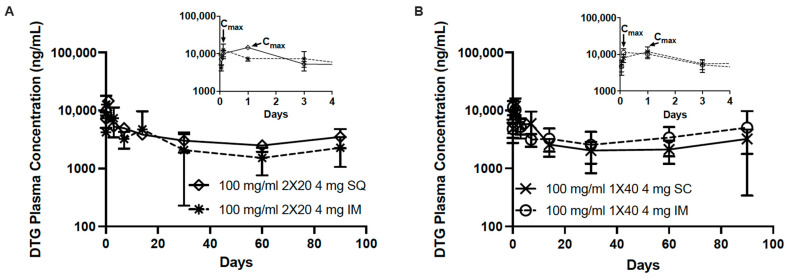
Comparing PK of 100 mg/mL DTG SQ and IM ISFIs with inlay graph through day 4 (**A**) 2 × 20 μL; 4 mg total dose (**B**) 1 × 40 μL; 4 mg total dose.

**Figure 7 pharmaceutics-14-00615-f007:**
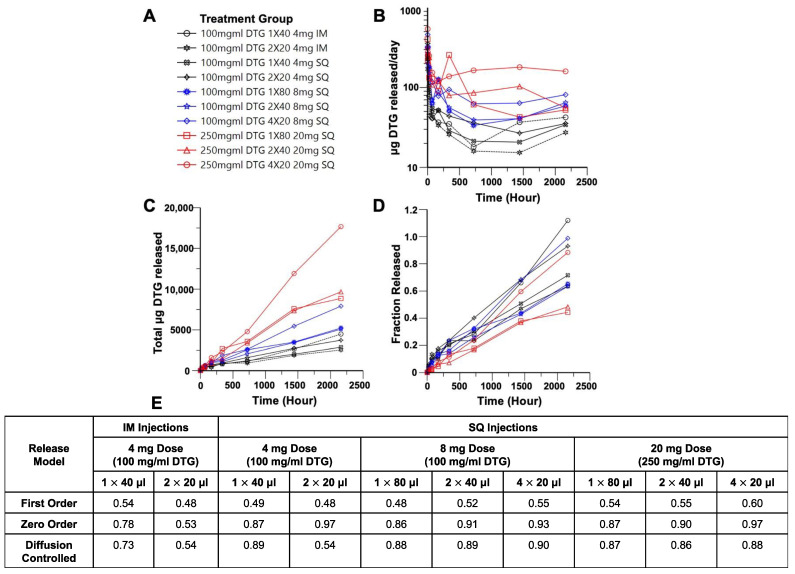
Median estimated deconvolution parameters of 4 × 20, 2 × 40, and 1 × 80 μL DTG-loaded ISFIs (1:2 *w*/*w* PLGA/NMP): (**A**) Legend for panels B–D. (**B**) Median estimated DTG release rate over 90 days. (**C**) Median estimated cumulative DTG mass over 90 days. (**D**) Median estimated fraction of loaded DTG dose over 90 days. (**E**) r^2^ values for mathematical models of first order, zero order, and diffusion-controlled (Higuchi model) release.

**Table 1 pharmaceutics-14-00615-t001:** Pharmacokinetic Non-compartmental Analysis.

Group	Dose (mg)	Median AUC_0–30d_ (h ∗ ng/mL)	25% Percentile	75% Percentile	*p*-Value	
100 mg/mL 1 × 40 IM	4	2,367,255.00	2,169,626.25	3,348,652.50	1.000	
100 mg/mL 2 × 20 IM	4	2,018,574.50	1,710,932.00	3,779,222.50		
100 mg/mL 1 × 40 SQ	4	2,692,192.50	2,223,818.25	3,814,503.75	1.000	
100 mg/mL 2 × 20 SQ	4	3,477,907.50	3,259,230.00	3,693,720.00		
100 mg/mL 1 × 80 SQ	8	5,302,942.50	4,170,672.50	6,184,407.50	1.000 0.911	0.993
100 mg/mL 2 × 40 SQ	8	4,462,922.50	3,750,026.25	5,408,896.25		
100 mg/mL 4 × 20 SQ	8	5,875,935.00	5,296,415.00	6,815,062.50		
250 mg/mL 1 × 80 SQ	20	7,941,840.00	6,143,650.00	12,325,440.00	1.000 0.923	0.945
250 mg/mL 2 × 40 SQ	20	7,461,075.00	6,121,780.00	8,639,070.00		
250 mg/mL 4 × 20 SQ	20	10,865,925.00	9,519,702.50	11917185.00		

## Data Availability

The data presented in this study are within the article and supplementary material, or on request from the corresponding authors.

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
