# Peer review of "Effects of Injection Volume and Route of Administration on Dolutegravir In Situ Forming Implant Pharmacokinetics"

_pharmaceutics, 2022, doi:10.3390/pharmaceutics14030615_

Round 1

Reviewer 1 Report

In this work, the authors reported an in-situ forming implant capable of delivering DTG in mice at a 4-fold higher concentration than the protein-regulated IC90 target, which could be administered via the SQ and IM routes and using different injection volumes and different numbers of injections. This work assessed the effects of route of administration, injection volume, and drug load on ISFI formation, degradation, and drug release in mice using ultrasound imaging and pharmacokinetics in mice. Authors did a very detailed study. However, there are some problems to be further improved as well:

  1. Abbreviations that appear for the first time in the manuscript should be marked in the abstract, such as SQ, DTG, IM, etc., which will affect the reader's understanding of the manuscript, please check.
  2. Figure 7 is not clear if I can't identify the different lines well. Please provide clearer pictures.
  3. In this study, the authors found no statistically significant difference in total drug exposure when the total dose administered was held constant regardless of route of administration or number of injections. However, the authors should evaluate the safety of different routes of administration and the number of injections.
  4. The authors need to recheck the manuscript very carefully and improve the language.

Author Response

Dear reviewer,

Thank you for the valuable feedback and recommendation. We have addressed all your comments to our best abilities and have included a point-by-point response to your comments in the attached response letter. 

Reviewer 2 Report

Referee Report

Title: Effects of injection volume & route of administration on dolutegravir in situ forming implant pharmacokinetics

Manuscript ID: pharmaceutics-1626632

By Joiner et al

Submitted to Pharmaceutics (ISSN 1999-4923)

Comment

This work investigated the impact of administration route, injection volume, and drug loading on ISFI formation, degradation, and drug release in a preclinical model using US imaging and pharmacokinetics. Based on the measurement, the authors found that when total administered dose is held constant, there is no statistical difference in total drug exposure regardless of the route of administration or number of injections. This is a very good piece of work and I only have some minor comments:

  1. Section 3: What frequency of US was used in the measurement?

  1. 3: According to Fig. 1A, the day length should be 1, 3, 5 … However, in Fig. 2 and 3, the US images demonstrated day 1, 2, 5 …

  1. Can the authors explain why only Fig. 4D has p < 0.001 when comparing IM and SQ?

  1. 5A and Fig. 5B: Figure caption missing information of the insets.

  1. Table 1: information of P-value does not match to other rows.

  1. 7A and Fig. 7B are too small to view.

Author Response

(The authors gave the same response as above.)

Round 2

Reviewer 2 Report

I am satisfied with the modification and correction done by the authors as per my comments. I accepted the authors' responses regarding my question. The presentation and quality of this work are improved.